# A metal-free photoactive nitrogen-doped carbon nanosolenoid with broad absorption in visible region for efficient photocatalysis

Yu Zhou[1,2,5], Xinyu Zhang[2,5], Guan Sheng[3,5], Shengda Wang[2], Muqing Chen ®[1] ✉, Guilin Zhuang ®[4], Yihan Zhu ®[3] ✉ & Pingwu Du ®[2] ✉

Riemann surfaces inspired chemists to design and synthesize such multi-dimensional curved carbon architectures. It has been predicted that carbon nanosolenoid materials with Riemann surfaces have unique structures and novel physical properties. Here we report the first synthesis of a nitrogen-doped carbon nanosolenoid (**N-CNS**) using bottom-up approach with a well-defined structure. **N-CNS** was obtained by a rational Suzuki polymerization, followed by oxidative cyclodehydrogenation. The successful synthesis of **N-CNS** was fully characterized by GPC, FTIR, solid-state $^{13}$C NMR and Raman techniques. The intrinsic single-strand molecular structures of **N-CNS** helices can be clearly resolved using low-dose integrated differential phase contrast scanning transmission electron microscopy (iDPC-STEM) technique. Possessing unique structural and physical properties, this long π-extended polymer **N-CNS** can provide new insight towards bottom-up syntheses of curved nanoribbons and potential applications as a metal-free photocatalyst for visible-light-driven $H_2$ evolution and highly efficient photocatalyst for photo-redox organic transformations.

In recent years, syntheses of nonplanar polycyclic aromatic hydrocarbons (PAHs) have attracted increasing attention due to their unique structures and novel physical properties[1–5]. Nonplanar carbon-based π-conjugated systems can be constructed with different topologies of bowls, hoops, and saddles, which can be considered as conjugated segments of larger 0D to 3D systems such as fullerenes[6–9], carbon nanotubes[10,11], helical nanocarbons[12–15], and porous graphene sheets[16,17]. Riemann surfaces are curved helical structures in chemistry, described as deformed versions of the complex plane in mathematics. Carbon nanosolenoids can be considered to follow a Riemann surface[18], in which one atomic carbon layer continuously spirals around the line perpendicular to the basal plane. Interestingly, Riemann surfaces were proposed as well-known objects in complex analysis and inspired scientists to design multidimensional carbon architectures (Fig. 1a)[19]. Some nonplanar structures were predicted to have many interesting mechanical, electronic, and magnetic properties[20,21].

Narrow strips of graphene with abundant edges and high aspect ratios, called graphene nanoribbons (GNRs), have great potential applications in photonics, electronics, and energy science[22–24]. Using a top-down method, GNRs can be synthesized by breaking apart graphene or carbon nanotubes. However, it is difficult to atomically

[1]School of Materials Science and Engineering, Dongguan University of Technology, 523808 Dongguan, Guangdong Province, China. [2]Key Laboratory of Precision and Intelligent Chemistry, Anhui Laboratory of Advanced Photon Science and Technology, Department of Materials Science and Engineering, University of Science and Technology of China, 96 Jinzhai Road, 230026 Hefei, Anhui Province, China. [3]Center for Electron Microscopy, Institute for Frontier and Interdisciplinary Sciences, State Key Laboratory Breeding Base of Green Chemistry Synthesis Technology, College of Chemical Engineering, Zhejiang University of Technology, 18 Chaowang Road, 310014 Hangzhou, Zhejiang Province, China. [4]College of Chemical Engineering, Zhejiang University of Technology, 18 Chaowang Road, 310014 Hangzhou, Zhejiang Province, China. [5]These authors contributed equally: Yu Zhou, Xinyu Zhang, Guan Shen. ✉e-mail: mqchen@ustc.edu.cn; yihanzhu@zjut.edu.cn; dupingwu@ustc.edu.cn

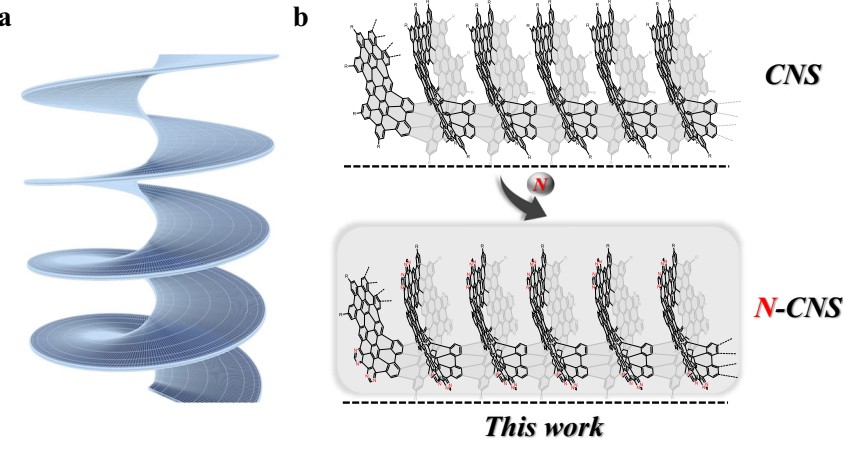

**Fig. 1 | Schematic illustrations of Riemann surface and nitrogen-doped carbon nanosolenoids (N-CNS). a** An example of Riemann's surface. **b** Design of fully π-extended **N-CNS**.

precise control over the GNR structure and further investigate their applications. Therefore, bottom-up synthesis seems to be a feasible approach to solve this problem. Well-defined GNRs can be constructed by surface-assisted or solution-based polymerization of precursors followed by cyclodehydrogenation reactions[25–27]. Among them, donor–acceptor (D–A) heterostructures can be fabricated through heteroatom doping in graphene strips with nitrogen, sulfur, and boron atoms at edges[28–31]. Owing to its electron-donating character and tendency to induce robust ferromagnetism in graphene at a high doping concentration, nitrogen has become the major heteroatom for the synthesis of heterostructures[32,33]. The electronic properties of GNRs could also be tuned by doping with nitrogen, which has been studied both theoretically and experimentally[34,35]. Fasel, Müllen, and coworkers used a stepwise cyclization/dehydrogenation sequence on a metal Au(111) surface to obtain nitrogen-doped graphene nanoribbons (N-GNRs), demonstrating a p–n junction character with an electric field of $2 \times 10^8$ V m$^{-1}$ and a band shift of 0.5 eV at the heterojunction[36]. Moreover, the synthesis of porous N-GNRs was reported by the Meyer group via Ullmann coupling on an Ag(111) surface[37]. However, surface-assisted synthesis strategies are difficult to scale up and make non-planar 3D strips. Very recently, we developed a facile solution-based strategy to synthesize all-carbon nanosolenoid material with Riemann surface[18]. Nanostructured graphitic carbon materials resembling a Riemann surface with helicoid topology are predicted to have interesting magnetic, electrical, and photonic properties[19–21]. For example, in 2016, Yakobson theoretically predicted a carbon solenoid with Riemann surfaces. This material could emerge a large magnetic field when a voltage is applied and become a brand new magnetic material[19]. The following experimental results proved that such carbon nanosolenoid material has magnetic properties[18]. Based on these previous results, we speculate that nitrogen-doped carbon nanosolenoid (N-CNS) heterojunction material with Riemann surfaces will possess special optoelectronic and photocatalytic properties, which is unexplored in this field. However, there is no report on the large-scale fabrication of nitrogen-doped carbon nanosolenoid (N-CNS) heterojunction material with Riemann surfaces.

Herein, we report the first synthesis and characterization of such a nitrogen-doped carbon material with interesting physical properties (Fig. 1b). The photophysical properties were investigated using UV–Vis absorption and fluorescence spectroscopy. Furthermore, the broad absorption of **N-CNS** in the visible region enables it as an excellent candidate for photocatalysis. The photocatalysis pathway for solar energy conversion helps to meet the growing demand for energy issues and environmental restoration[38,39]. Currently, efficient and environmentally friendly photocatalysts are in high demand for many important energy conversion applications, such as solar water splitting, photocatalytic organic reactions, and so on[40–42]. In addition, metal-free catalysts are also of great significance in green chemistry and attract much attention in research[43,44]. Metal-free catalysts have many advantages that include low cost and ready availability, low toxicity, higher stability in air and water, and increased synthetic efficiency due to the avoidance of the time-consuming removal of toxic metal traces. **N-CNS**, which does not contain any metal component and has photocatalytic properties under visible light, can be used as a potential metal-free catalyst for both visible light-driven $H_2$ production and efficient photocatalytic organic transformations.

## Results

### Molecular design of N-CNS

Figure 2 shows the conceptual structure with helicoid topology, in which nanostructured N-doped carbon nanosolenoid material resembles a Riemann surface. The appropriate building units are rationally chosen to obtain the desired structural features. A hexaphenylbenzene (HPB) derivative was chosen as the PAH building unit to achieve the large π-extended feature of **N-CNS**. To obtain the heterostructure of **N-CNS**, 5-bromopyrimidine serves as the starting material for introducing nitrogen atoms and Sonogashira coupling can be performed to produce 1,2-di(pyrimidin-5-yl)ethyne. Next, the Diels–Alder reaction between 5,10-dibromo-1,3-diphenyl-2H-cyclopenta[l]phenanthren-2-one and 1,2-di(pyrimidin-5-yl)ethyne can afford the HPB derivative unit **M1**. Another building unit **M2** was designed as the linker with two *ortho*-boryl groups to achieve the helicoid topology of **N-CNS**. Two *ortho*-*tert*-butyl groups in **M2** were introduced to improve the solubility of precursor **P1** and **N-CNS**.

### Solution synthesis of N-CNS with large lateral π-extension

The synthesis procedure of the **N-CNS** is summarized in Fig. 2. The essential precursor **P1** was fabricated by combining the molecular building blocks 5,5′-(6,11-dibromo-1,4-diphenyltriphenylene-2,3-diyl)dipyrimidine (**M1**) and 2,2′-(4,4″-di-*tert*-butyl-[1,1′:2′,1″-terphenyl]-4′,5′-diyl)bis(4,4,5,5-tetramethyl-1,3,2-dioxaborolane) (**M2**). Then, **P1** was further performed Scholl reaction to obtain large π-conjugated **N-CNS**. The detailed synthesis procedure of these molecular building blocks is described in Supplementary Information. The polymerization of **M1** and **M2** was performed by Suzuki–Miyaura coupling reaction using Pd(PPh$_3$)$_4$ as the catalyst, toluene and water as solvents, and reacted at 110 °C for 60 h to obtain the essential precursor (**P1**) with a yield of 83%. Finally, **N-CNS** was obtained by oxidative

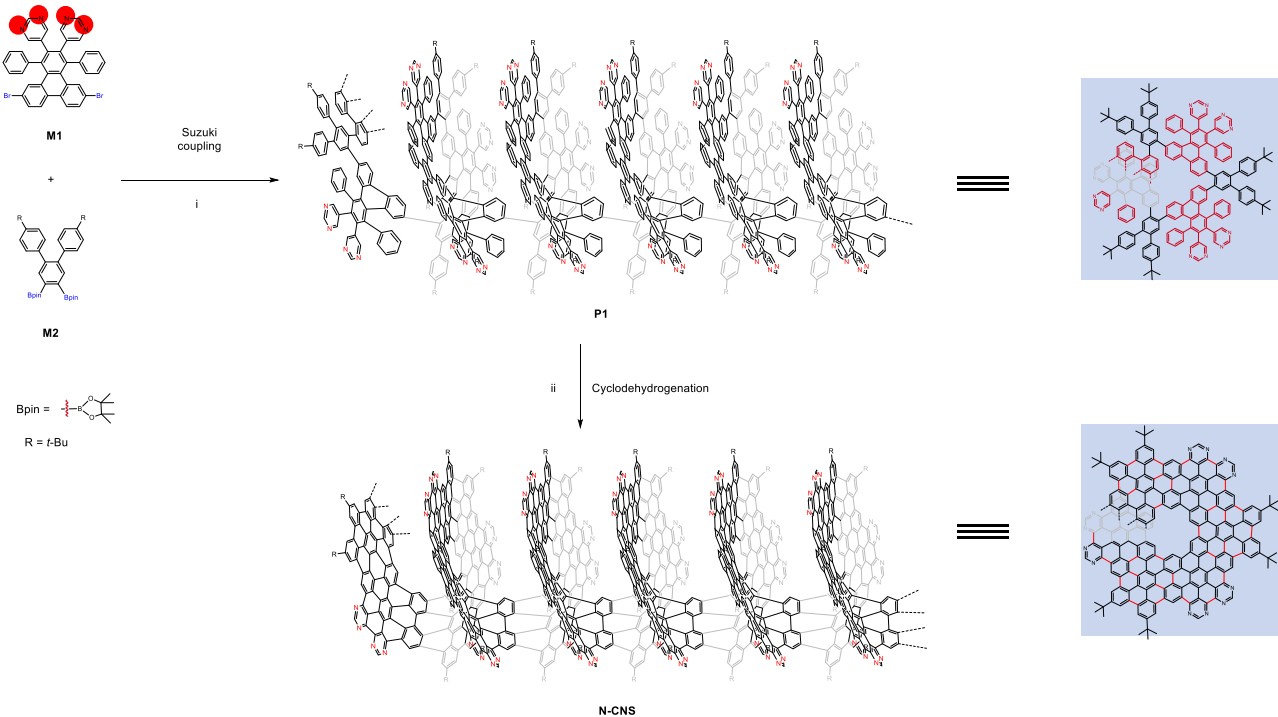

**Fig. 2 | Synthetic approach to N-CNS.** Reagents and conditions: (i) **M1** (1.0 equiv.), **M2** (1.0 equiv.), K$_2$CO$_3$ (10 equiv.), Aliquat 336 (5 mol%), Pd(PPh$_3$)$_4$ (10 mol%), Ar, toluene/H$_2$O ($v/v$, 5:1), 110 °C, 60 h; (ii) **P1** (1.0 equiv.), DDQ (16 equiv.), TfOH, Ar, anhydrous CH$_2$Cl$_2$, 0 °C, 24 h.

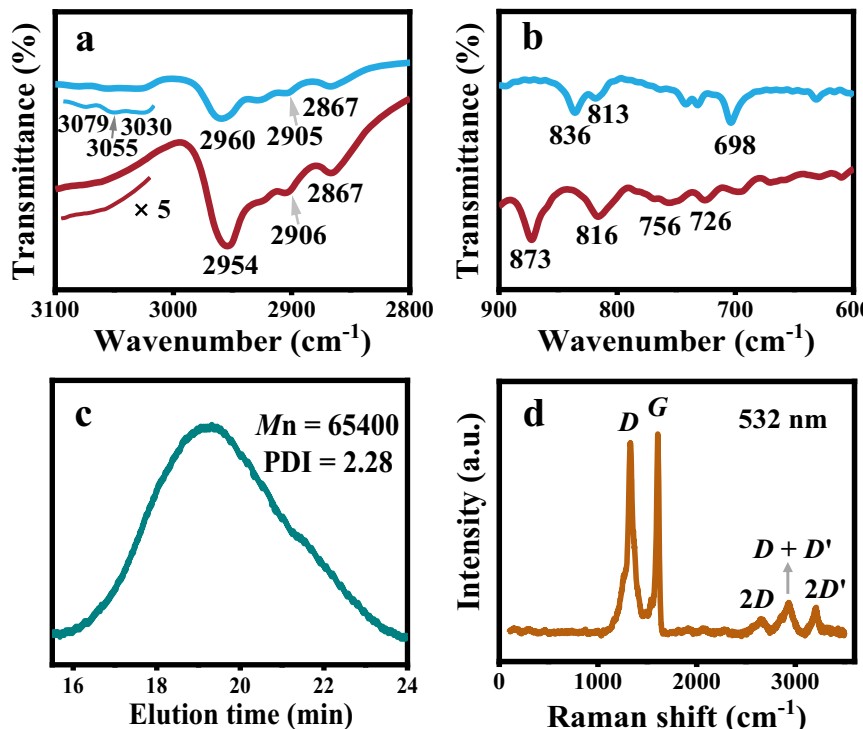

**Fig. 3 | Spectroscopic characterization of N-CNS. a** and **b** Representative FTIR spectral regions of precursor **P1** (blue lines) and **N-CNS** (red lines). **c** GPC trace of **N-CNS**. **d** Raman spectrum of **N-CNS** measured at 532 nm (2.33 eV) on a powder sample.

cyclodehydrogenation of **P1** using 2,3-dichloro-5,6-dicyano-1,4-benzoquinone and trifluoromethanesulfonic acid at 0 °C for 24 h. During the reaction, we found that the solution color quickly turned black, and finally, a dark purple solid was obtained. Moreover, the fluorescence of the solution was changed from blue to red, further

confirming the increasing degree of π-extended conjugation of the backbone.

Gel permeation chromatography (GPC) was used to measure the average molecular weight of **N-CNS** (Fig. 3c). The weight average molecular weight ($M_W$), relative number-average molecular weight

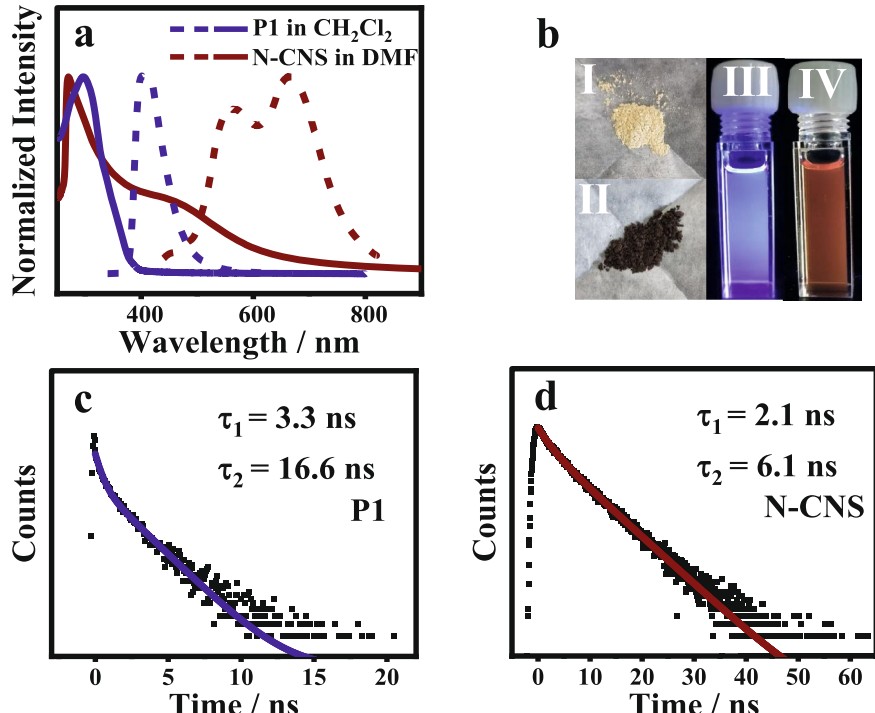

**Fig. 4 | Photophysical properties of N-CNS. a** UV–Vis absorption (dots) and fluorescence spectra (lines) of **N-CNS** (wine) in DMF and precursor **P1** (blue) in CH$_2$Cl$_2$. **b** Solid powder of precursor **P1** (I) and **N-CNS** (II), precursor **P1** in CH$_2$Cl$_2$ solution (III) and **N-CNS** in DMF solution (IV) under UV irradiation at 365 nm. **c**, **d** Emission lifetimes for precursor **P1** in CH$_2$Cl$_2$ (black dots: data; blue line: fitting) and **N-CNS** in DMF (black dots: data; wine line: fitting).

($M_n$), and polydispersity index (PDI) of **N-CNS** were obtained. The molecular weight distribution of **N-CNS** shows a single broad peak with a PDI of 2.28. The $M_n$ of **N-CNS** is 65,400 g mol$^{-1}$. Considering that one helical pitch consists of three monomeric units, this $M_n$ corresponds to ~76 monomer units and contains ~25 spiral pitches.

## Bulk spectroscopic characterizations of N-CNS

The highly efficient cyclodehydrogenation of precursor **P1** into **N-CNS** was confirmed by Fourier transform infrared (FTIR), Raman spectroscopy, and solid-state $^{13}$C NMR measurements. FTIR analysis of the material **N-CNS** after the cyclodehydrogenation revealed that the signals at 3030, 3055, and 3079 cm$^{-1}$ originating from aromatic C–H stretching vibrations in **P1** were diminished[26]. Similar peaks (2906 and 2867 cm$^{-1}$ for **N-CNS**, 2905 and 2867 cm$^{-1}$ for **P1**) associated with the C–H stretching of alkyl chains were observed upon cyclodehydrogenation, demonstrating the integrity of the alkyl substituents. The vibration bands at 698, 813, and 838 cm$^{-1}$ from mono- and disubstituted benzene rings were attenuated (Figs. 3a, b, S2 and S3)[45]. Furthermore, the appearance of a new band maximized at 873 cm$^{-1}$ confirmed the *opla* band which is a typical signal for aromatic C–H at the cove position (Fig. 3b)[45–47]. All these observations indicate the highly efficient conversion of precursor **P1** into **N-CNS**.

As a useful tool for characterizing carbon nanomaterials, Raman spectroscopy was performed which is sensitive to the structural and edge defects to confirm the high structural quality of **N-CNS**. The result shows the typical structural feature for graphene nanoribbons with characteristic intense G and D bands[25]. The Raman spectrum of **N-CNS** (excited at 532 nm, powder sample) demonstrates the typical G-band peak and D-band peak for armchair-type GNRs observed at 1604 and 1329 cm$^{-1}$, respectively (Fig. 3d), which is consistent with those of GNRs obtained by bottom-up synthesis[48]. Double formants at 2649, 2934, and 3206 cm$^{-1}$ were initially assigned to 2D, D + D′, and 2D′ peaks, respectively. The D-band peak can be explained by collective modes of the confined hexagonal rings for the confinement of π-electronics into a finite-size domain in large π-extended PAHs[49].

Solid-state $^{13}$C NMR spectra were performed to confirm the structural changes before and after the Scholl reaction (Fig. S4). The characteristic single peak ($\delta = 31.28$ ppm for **P1** and 31.92 ppm for **N-CNS**) can be assigned to *t*-butyl carbons. **P1** shows multiple peaks ($\delta = 109.13$–162.02 ppm), which can be assigned to the aromatic carbons. The main aromatic $^{13}$C NMR signals of **N-CNS** centered at ~124 ppm, probably caused by main π-conjugated carbon atoms in the **N-CNS**. Moreover, the peaks of **N-CNS** in the aromatic zone move slightly to the lower field because of the increasing π-conjugation.

## Physical properties

The photophysical properties of **N-CNS** and **P1** were characterized in solution by steady-state spectroscopy and time-resolved spectroscopy (Fig. 4). The absorption spectrum of **P1** only shows a narrow absorption band at 250–380 nm maximized at 300 nm. In sharp contrast, the UV–Vis spectrum of **N-CNS** in DMF showed a broad redshifted absorption band in the range of 260–800 nm maximized at ~500 nm, indicating that a large π-conjugated structure was formed after cyclodehydrogenation. Furthermore, under an excitation at 350 nm, **N-CNS** exhibits a broad emission band at 440–820 nm maximized at 564 and 661 nm, demonstrating significant redshift compared to the reference precursor **P1** (maximized at 413 nm). This emission redshift is consistent with the UV–Vis result, indicating a large and well-extended conjugation nature in **N-CNS**. As shown in Fig. 4b, the precursor **P1** is a light-yellow solid, while **N-CNS** becomes a black solid. For comparison, the dilute DMF solution of **N-CNS** showed an intense red photoluminescence under irradiation by a hand-held UV lamp at $\lambda = 365$ nm, while the precursor **P1** in solution presented an intense blue photoluminescence.

Moreover, the luminescence lifetime ($\tau_s$) of precursor **N-CNS** and **P1** were measured by time-resolved fluorescence decay using the time-resolved photoluminescence (TRPL) technique (Figs. 4c, d). The luminescence lifetime of **N-CNS** follows second-order kinetics with lifetimes of 6.12 and 2.13 ns at 661 nm when excited at ~420 nm. For comparison, **P1** shows longer fluorescence lifetimes at 16.64 and

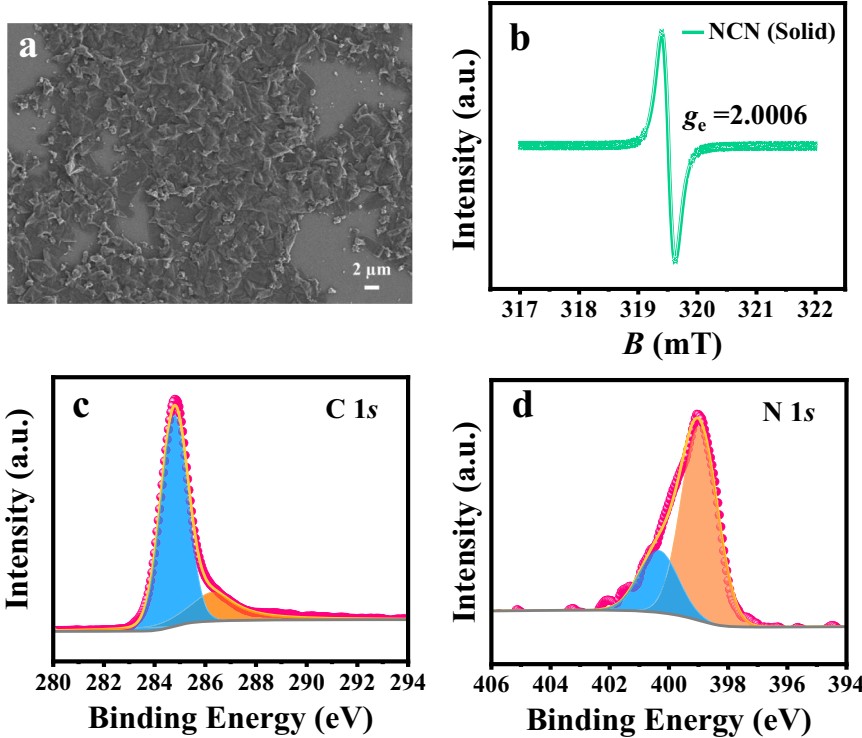

**Fig. 5 | SEM, AFM, room temperature EPR spectrum and XPS spectra characterizations of N-CNS. a** SEM image of bulk **N-CNS**, scale bar 4 μm; inset, AFM image of **N-CNS**, scale bar 6 Å. **b** Room temperature EPR spectrum of solid powder. **c** C 1*s* and **d** N 1*s* XPS spectra of **N-CNS**.

3.28 ns at 413 nm when excited at ~320 nm. The different emission lifetimes probably result from the different excited states of **P1** and **N-CNS**.

The morphology and microstructures of **N-CNS** were further investigated by scanning electron microscopy (SEM) and atomic force microscope (AFM). As shown in Fig. 5a, bulk **N-CNS** has layered structures consisting of thin sheets with micrometer-long wrinkles. AFM image showed that **N-CNS** can self-assemble into well-defined helix bundles on the surface (inset, Fig. 5a), which is consistent with our previous study.[18] To further investigate the electronic properties of **N-CNS**, EPR measurements were carried out using a solid-state sample at room temperature (Fig. 5b). **N-CNS** displayed a typical single EPR peak, with a $g_e$ value of ~2.0006, which confirms unpaired electrons and magnetic properties exist in this present solenoid material[18,19]. Moreover, the chemical compositions and states of **N-CNS** were investigated by X-ray photoelectron spectroscopy (XPS). In Fig. 5c, C 1*s* spectrum of **N-CNS** shows two peaks at 284.8 and 286.3 eV, corresponding to $sp^2$ bonded carbon–carbon, and $sp^2$ bonded carbon-nitrogen (N = C–N), respectively. Figure 5d exhibits N 1*s* XPS spectra, the peak at 399.5 eV could correspond to $sp^2$ N bonding of (C = N–C). Ni 2*p*, Co 2*p*, Fe 2*p*, and Pd 3*d* XPS spectra of **N-CNS** are shown in Fig. S8. These results indicate that all metal elements involved in the synthetic reactions are absent from the samples, further demonstrating that **N-CNS** is a metal-free carbon material.

The explicit real-space structural elucidation of such organic molecules requires direct high-resolution structural imaging using electron microscopy. However, these strands of **N-CNS** organic molecules are extremely vulnerable to electron beam irradiation and thus prohibit traditional electron microscopy techniques from imaging the integral structures. The recent advances in low-dose electron microscopy allow the direct imaging of beam-sensitive materials[50–52], including metal-organic frameworks[3,52–54], covalent-organic framework[52,55,56] and organic molecules[55]. With the development of low-dose electron microscopy methods, integrated differential phase contrast scanning transmission electron microscopy (iDPC-STEM)

shows great advantages in the imaging of low-*Z* elements and electron beam-sensitive materials[18,57,58]. Accordingly, we tried to image the organic **N-CNS** helices using the low-dose iDPC-STEM technique to maintain the structural integrity of these helices. Although the strands of **N-CNS** helices tend to bundle together due to the strong intermolecular interactions, prolonged probe sonication allows the segregation of a considerable fraction of single-strand **N-CNS** helices that can be clearly identified and elucidated by low-dose iDPC-STEM imaging. As we can see in Fig. 6, the intrinsic single-strand molecular structures of **N-CNS** helices can be clearly resolved from both [100] and [201] projections of the proposed 1D periodic structural model (Fig. S5). Figure 6a shows the iDPC-STEM image of a single-strand **N-CNS** helix projected along [100] axis, which exhibits fringe contrast characteristics for the periodic helical structures. The measured fringe spacing (*b*′) of the **N-CNS** helix is 0.40 ± 0.03 nm and the helical width (*d*′) is 2.4 ± 0.2 nm, respectively, which matches well with the proposed molecular structural model of **N-CNS** helix. As shown in Figs. 6c and S7, the calculated helical width of *d* is 25.12 Å (without the consideration of *tert*-butyl) and the fringe spacing of *b* is 4.110 Å; then the resulting angle γ is 2.974° (γ = argtan (*b*/(*d*\**π*))). It has been widely acknowledged that the iDPC-STEM contrast closely resembles the electrostatic potential of the chemical structure[18,59]. The present results demonstrate the iDPC-STEM image contrast closely resembles the simulated projected potential map of the proposed **N-CNS** helix model, as shown in Figs. 6b and c. Moreover, the helical structural model is further validated from another iDPC-STEM image taken along the [201] direction of the structural model (Fig. 6d), from which the **N-CNS** helix is tilted away from the image plane as shown in Fig. 6f. The STEM image exhibits arrays of bright dots that corresponding to the projected columns of benzene rings in the **N-CNS** helix, which match well with the simulated projected potential map as shown in Fig. 6e. Taken together, low-dose iDPC-STEM imaging along both flat-lying and tilted projections of single-strand **N-CNS** helices unambiguously confirms the proposed molecular structural mode of nitrogen-doped carbon nanosolenoid with Riemann Surfaces.

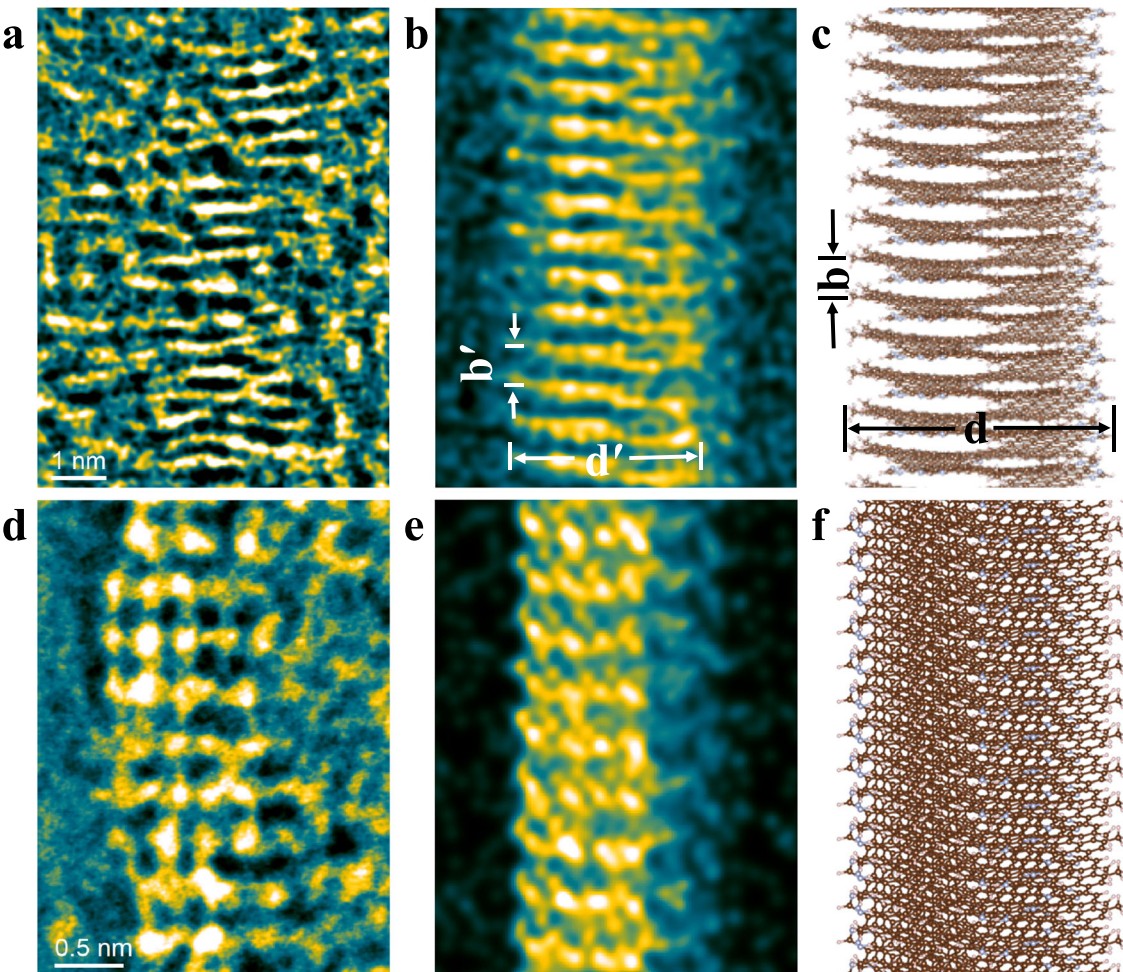

**Fig. 6 | iDPC-STEM characterizations of N-CNS.** Low-dose iDPC-STEM images (**a**, **d**), simulated projected potentials (**b**, **e**), and structural models (**c**, **f**) of single-strand **N-CNS** along [100] (upper figures) and [201] (lower figures) axes respectively. The false-colored images are rendered in aquatic color code. The structural model used for simulating the electrostatic potential is constructed by adhering the single-strand **N-CNS** along a specific projection with a properly situated amorphous carbon layer with a thickness of 1 nm. A specific point-spread-function (PSF) width of 1.6 Å was used for the simulation.

## Photocatalytic properties

Since **N-CNS** has excellent photophysical properties, it could be a potential metal-free photocatalyst for energy conversion. The photocatalytic $H_2$ evolution performance using **N-CNS** was further evaluated under the light irradiation of a 300 W Xe lamp equipped with a UV cut-off filter ($\lambda > 420$ nm). The typical photocatalytic $H_2$ production of **N-CNS** is schematically illustrated in Fig. 7a. Impressively, **N-CNS** can be used as a metal-free photocatalyst for hydrogen production in water. Figure 7b shows the photocatalytic activity of **N-CNS** for $H_2$ production in the presence of electron donors. With increasing concentrations of the electron donors ($Na_2S$ and $Na_2SO_3$), the photocatalytic $H_2$ evolution rate rapidly increased and reached a maximum value of $190.2 \pm 21.2$ μmol $g^{-1}$ $h^{-1}$ in the presence of 0.75 M $Na_2S$ and 1.05 M $Na_2SO_3$. When the concentrations of $Na_2S$ and $Na_2SO_3$ increased to 1.0 and 1.4 M, respectively, the $H_2$ evolution rate decreased to $182.8 \pm 17.5$ μmol $g^{-1}$ $h^{-1}$, which is probably due to the loss of visible light energy hindered by the undissolved $Na_2SO_3$. Compared to **N-CNS**, the $H_2$ evolution rate of commercial photocatalyst $TiO_2$ P25 was only 5.5 μmol $g^{-1}$ $h^{-1}$ under the same conditions upon visible light irradiation (Table S2), indicating that **N-CNS** has better photocatalytic activity than P25 under visible light using the same reaction conditions. We also compared **N-CNS** with other metal-free materials reported in the literature for hydrogen evolution and these results are shown in Table S3. Most of these materials exhibited low $H_2$ evolution rates, and

only a few materials demonstrated rates higher than 100 μmol $g^{-1}$ $h^{-1}$. In contrast, the hydrogen evolution rate of **N-CNS** is 190 μmol $g^{-1}$ $h^{-1}$. This result demonstrated that **N-CNS** has good photocatalytic properties for $H_2$ production. Besides, the photocatalytic activity of **N-CNS** using different sacrificial electron donors was also measured for comparison (Fig. S9). As shown in Fig. 7d, the **N-CNS** photocatalyst exhibited excellent stability and maintained a similar photocatalytic activity for 36 h. An obvious transient photocurrent response was observed, suggesting the good photo-induced electron separation/transport property of **N-CNS** (Fig. S13). To confirm the electronic band positions of the as-prepared materials, Mott–Schottky (MS) measurements were carried out (Fig. S11). The conduction-band position is −0.46 eV (vs. NHE) for **N-CNS**. The absorption edge of **N-CNS** is located at 670 nm, corresponding to the band gap of 1.85 eV (Fig. S12). Combined with the band gap of **N-CNS**, the valence-band position can be determined as 1.39 eV (vs. NHE). All these results demonstrate that **N-CNS** can perform as a good metal-free photocatalyst for visible-light-driven $H_2$ evolution.

**N-CNS** can also be proposed as a novel photocatalyst to realize the C–H functionalization of 2-phenyl-1,2,3,4-tetrahydroisoquinoline (**1a**)[60–63]. Catalytic activity of **N-CNS** toward photocatalytic aza-Henry reaction was further evaluated with **1a** and nitromethane as starting materials. The reaction was carried out in the presence of $O_2$ with a 9 W blue LED as the light source. To our delight, the desired C–H

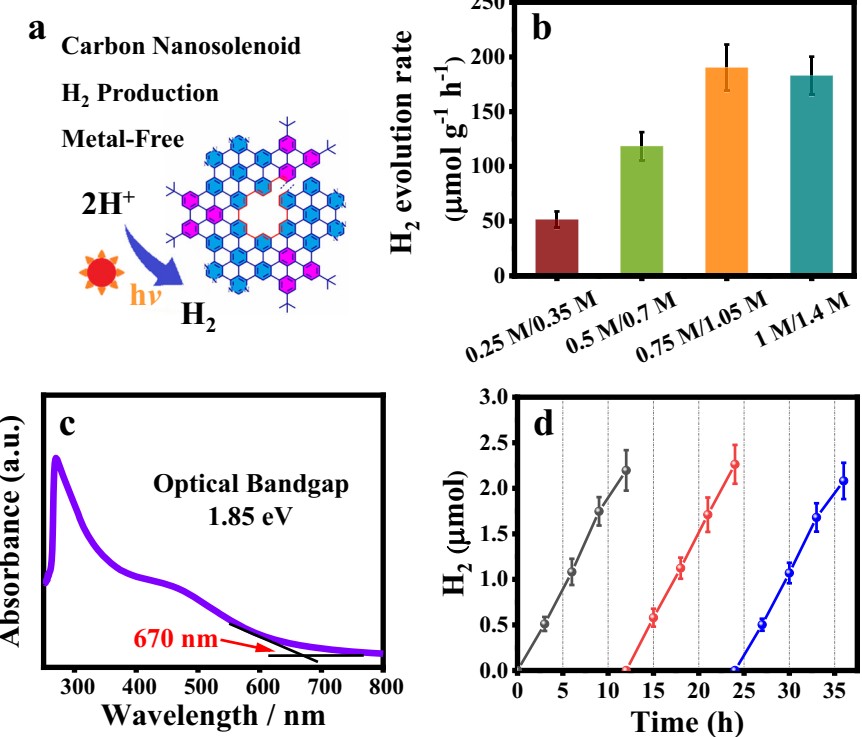

**Fig. 7 | Photocatalytic H₂ production by N-CNS. a** Schematic illustration of photocatalytic H₂ production by **N-CNS**. **b** H₂ evolution rate of **N-CNS** photocatalyst at different concentrations of the hole scavenger under visible light (wine: 0.25 M Na₂S and 0.35 M Na₂SO₃; green: 0.5 M Na₂S and 0.7 M Na₂SO₃; orange: 0.75 M Na₂S and 1.05 M Na₂SO₃; cyan: 1 M Na₂S and 1.4 M Na₂SO₃;). **c** UV–Vis absorption spectrum of **N-CNS** in DMF with calculated optical bandgap. **d** Cycling runs of **N-CNS** for photocatalytic H₂ evolution in the presence of 1.0 mg **N-CNS** photocatalyst in a 30 mL aqueous solution containing 0.75 M Na₂S and 1.05 M Na₂SO₃ (black: cycle 1; red: cycle 2; blue: cycle 3). The error bars in **b** and **d** show standard deviation (SD).

functionalization product **1b** was isolated in an excellent 95% yield by irradiation for 5 h with a catalytic amount of **N-CNS** (Table 1). And the use of visible light (300 W Xe lamp equipped with a UV cut-off filter, $\lambda > 420$ nm) resulted in a 90% yield. In sharp contrast, only a 9% yield of **1b** was obtained when the reaction was performed using the precursor **P1**. Control experiments revealed that catalyst, light, and oxygen are crucial for the formation of **1b**. Moreover, we also performed the absorption spectrum to investigate the stability of **N-CNS** before and after the photocatalytic aza-Henry reaction (Fig. S14). After the reaction, a three-step procedure including centrifugation, separation, and washing was done and **N-CNS** was performed in a new cycle, no significant loss of photocatalytic activity was observed after 10 cycles of 12 h (Fig. S22).

Besides the easy functionalization of tetrahydroisoquinoline, various other substituents were also subjected to photocatalytic organic transformations using **N-CNS** as the photocatalyst. As shown in Table 2, the catalytic results are also excellent for the benzylamine homocoupling reaction and oxygenation reaction of thioanisoles with all the yields >90%. Under light irradiation, the benzylamine homocoupling produced the product of N-benzyl-1-phenylmethanimine (**3a**) in a yield of 90.1%, as checked by ¹H NMR. Oxygenation reaction of methyl(phenyl)sulfane was performed under similar photocatalytic reaction conditions and the product of (methylsulfinyl)benzene (**3b**) had a yield as high as 94.9% in only 5 h. In addition, more sulfinylbenzene compounds with the amendment of different groups at the *para*-position, such as –OMe (**2c**), –F (**2d**), were examined. The yields of 1-methoxy-4-(methylsulfinyl)benzene (**3c**) and 1-fluoro-4-(methylsulfinyl)benzene (**3d**) were calcualted to be 90.0% and 90.9%, respectively. All the high yields indicated that the **N-CNS** is an excellent metal-free photocatalyst in many photoredox reactions.

## Discussion

In summary, we reported a facile bottom-up synthesis of a large well-extended nitrogen-doped carbon nanosolenoid (**N-CNS**) with Riemann surface by a Pd-mediated Suzuki–Miyaura coupling followed by a Scholl reaction for cyclodehydrogenation. **N-CNS** was fully characterized by GPC, FTIR, Raman, solid-state ¹³C NMR spectra, and iDPC-STEM techniques. Its unique photophysical properties were investigated using UV–Vis, fluorescence, TRPL spectroscopy. Notably, **N-CNS** can be used as a simple polymeric metal-free photocatalyst for hydrogen production in water and a highly efficient photocatalyst such as aza-Henry reactions and oxygenation reactions of thioanisoles in the absence of any noble metals. The present results can shed light on the synthesis of novel carbon nanomaterials and explore their applications in solar energy conversion.

## Methods

### Compounds preparation
See Supplementary Information for details of the syntheses and characterization of **M1** and **M2**.

### Synthesis of P1
To a degassed suspension of 5,5'-(6,11-dibromo-1,4-diphenyl-triphenylene-2,3-diyl)dipyrimidine (**M1**) (234 mg, 0.336 mmol), 2,2'-(4,4"-di-*tert*-butyl-[1,1':2',1"-terphenyl]-4',5'-diyl)bis(4,4,5,5-tetra-methyl-1,3,2-dioxaborolane) (**M2**) (200 mg. 0.336 mmol), potassium carbonate (465 mg, 3.36 mmol), Aliquat 336 (7.0 mg, 0.017 mmol, 5 mol%) in toluene (5 mL) and H₂O (1 mL) was added Pd(PPh₃)₄ (39.3 mg, 0.034 mmol), then the mixture was degassed for 15 min. The mixture was then heated at 110 °C for 60 h under a nitrogen atmosphere. Upon the reaction cooling to room temperature, the mixture was diluted with 1 M aqueous hydrochloric acid, extracted with CH₂Cl₂,

**Table 1 | The photocatalytic aza-Henry reactions[a]**

| Variation | Yield of 1b (%)[b] |
| --- | --- |
| No photocatalyst | Trace |
| No light | Trace |
| No $O_2$ | 22.2 |
| **P1** | 9.1 |
| **N-CNS** with visible light | 90.0 |
| **N-CNS** | 95.1 |

[a]Reaction conditions: 2-phenyl-1,2,3,4-tetrahydroisoquinoline **1a** (0.5 mmol), photocatalyst (1.5 mg), $CH_3NO_2$ (3 mL), $O_2$ (1 atm), LED lamp (9 W).
[b]Isolated yield.

and concentrated under vacuum. The resulting precipitate was washed intensively with 0.5 M HCl solution in water, filtered, and washed with water, methanol, acetone, methanol, and hexane. The precipitate was collected, giving 304 mg of polymer **P1** as a pale yellow solid (yield: 83%).

## Synthesis of N-CNS

After a solution of polymer **P1** (500 mg, 0.55 mmol), 2,3-dichloro-5,6-dicyanobenzoquinone (DDQ:2.01 g, 8.9 mmol) in anhydrous DCM (50 ml) was degassed with argon for 20 min, trifluoromethanesulfonic acid (3.0 mL) was added to the mixture. Then the reaction mixture was stirred at 0 °C for another 24 h under an argon atmosphere and quenched with saturated $NaHCO_3$ solution, filtered, and washed with methanol, acetone, $CH_2Cl_2$, and methanol. The precipitate was collected to obtain the title polymer **N-CNS** as a dark black solid (yield: 76%): $M_n = 65{,}391$ g mol$^{-1}$ and $M_w = 149{,}204$ g mol$^{-1}$ by GPC.

## Aza-Henry reaction

To a flame-dried 10 mL vial equipped with a magnetic stir bar with **1a** (104.6 mg, 0.5 mmol) and **N-CNS** (1.5 mg). Nitromethane solvent (3 mL) was transferred to the vial via syringe under oxygen and stirred at a blue LED lamp (9 W) for 5 h. After the reaction, the **N-CNS** was separated by centrifugation and washed with DCM which could be used for the catalyzer again. The crude product **1b** was in the solvent and purified by silica gel chromatography to afford the desired **1b** as a yellow oil in a yield of 95% (127.5 mg). $^1$H NMR ($CDCl_3$, 400 MHz): δ 7.33–7.16 (m, 5H), 7.13 (d, $J$ = 7.2 Hz, 1H), 6.98 (d, J = 8.2 Hz, 2H), 6.85 (t, J = 7.3 Hz, 1H), 5.55 (t, J = 7.2 Hz, 1H), 4.87 (dd, J = 11.8, 7.8 Hz, 1H), 4.56 (dd, J = 11.8, 6.6 Hz, 1H), 3.71–3.56 (m, 2H), 3.14–3.04 (m, 1H), 2.84–2.75 (m, 1H).

The recycling process of **N-CNS** photocatalytic for aza-Henry reaction: To a flame-dried 10 mL vial equipped with a magnetic stir bar with **1a** (104.6 mg, 0.5 mmol) and recycling **N-CNS** (1.5 mg). Nitromethane as the solvent (3 mL) was transferred into the vial via syringe under oxygen and stirred under a blue LED lamp (9 W) for 5 h. Then, the **N-CNS** catalyst was recovered for another cycle. After a few new cycles, the product **1b** can be still obtained with a similar yield.

## Benzylamine homocoupling reaction

To a flame-dried 10 mL vial equipped with a magnetic stir bar with **2a** (1 mmol) and **N-CNS** (1.5 mg). MeCN (3 mL) was transferred to the vial via syringe under oxygen and stirred at a blue LED lamp (9 W) for 5 h. The mixture was filtered to remove the catalyst to get main product **3a**. The yields were determined by $^1$H NMR analysis (Fig. S18).

## Photocatalytic oxidation of thioethers

To a flame-dried 10 mL vial equipped with a magnetic stir bar with **2b–2d** (1 mmol) and **N-CNS** (1.5 mg). MeOH (3 mL) was transferred to the vial via syringe under oxygen and stirred at a blue LED lamp (9 W) for 5 h. The mixture was filtered to remove the catalyst to get the main product **3b–3d**. The yields were determined by $^1$H NMR analysis (Figs. S19–S21).

## Spectroscopic analysis

High-resolution mass spectrometry (HR-MS) analyses were carried out using MALDI-TOF-MS techniques. NMR spectra were recorded on Bruker BioSpin ($^1$H 400 MHz, $^{13}$C 100 MHz) spectrometer, and chemical shifts were reported as the delta scale in ppm relative to $CDCl_3$ (δ = 7.26 ppm) for $^1$H NMR and $CDCl_3$ (δ = 77.0 ppm) for $^{13}$C NMR. Data are reported as follows: chemical shift, multiplicity (s = singlet, d = doublet, t = triplet, m = multiplet, br = broad signal), coupling constant (Hz), and integration. UV–vis absorption spectra were performed on a UNIC-3802 spectrophotometer. Atomic force microscope (AFM) measurements were performed using a Dimension ICON microscope (Bruker) in the tapping mode in a clean room environment. All anhydrous solvents and starting chemical reagents for syntheses were purchased from commercial suppliers (Aldrich or Acros) and used without further purification unless otherwise noted. All moisture- or air-sensitive reactions were carried out in a dry reaction vessel under an inert atmosphere (argon or nitrogen) using standard Schlenk techniques. Preparative column chromatography was performed on silica gel (size 200–300 mesh). Analytical thin-layer chromatographies (TLC) were performed with silica gel HSGF 254. Flash chromatography was performed on silica gel (300–400 mesh). Gel permeation chromatography (GPC) was carried out using a G1316A PL gel column with a rate of 1.0 min/mL in DMF assured by G1310B Iso.pump and detected by a G1362A differential refractive index detector. The solid-state $^{13}$C NMR experiments were conducted on a Bruker AVANCE NEO600WB spectrometer ($^1$H 600.16 MHz, $^{13}$C 150.93 MHz) using a Bruker 3.2 mm H/X/Y probe. The samples were packed in 3.2 mm o.d. $ZrO_2$ rotos. Gly was used as an external reference to calibrate the

**Table 2 | The photocatalytic oxygenation of benzylamine and thioanisoles[a]**

| Product | Yields (%)[b] |
|---|---|
| 3a | 90.1 |
| 3b | 94.9 |
| 3c | 90.0 |
| 3d | 90.9 |

[a]Reaction conditions: benzylamine **2a** (0.5 mmol), photocatalyst (1.5 mg), MeCN (3 mL), $O_2$ (1 atm), blue LED lamp (9 W); thioanisole **2b–2d** (0.5 mmol), photocatalyst (1.5 mg), MeOH (3 mL), $O_2$ (1 atm), blue LED lamp (9 W).
[b]The yields were determined by $^1$H NMR.

radiofrequency (rf) field strength and chemical shift scale ($\delta$ ($^{13}$C) = 176.03 ppm).

## The iDPC-STEM imaging

The low-dose high-resolution iDPC-STEM images were obtained under a Cs-corrected electron microscope operated at 300 kV. The beam current was reduced to 1 pA, the convergence angle was 25 mrad and the collection angle of iDPC-STEM imaging was set to 7–29 mrad. The projected electrostatic potential was simulated using the QSTEM code with a point-spread-function (PSF) width of 1.5 Å over a single-strand CNS structural model embedded in a 1 nm-thick amorphous carbon layer.

## Computational details

**Geometrical structure and electronic structures of N-CNS.** Geometrical optimization of the whole cell was carried out utilizing the Vienna an initio simulation package (version: VASP 6.3.1)[64]. Perdew–Burke–Ernzerhof (PBE)[65] in terms of the gradient of electronic density described the exchange and correlation (XC) interaction in the Kohn–Sham equation. The interaction between ions and electrons was treated by the projector-augmented wave (PAW)[66]-based pseudopotential, featuring greater computational efficiency as well as high accuracy. Specifically, outer electrons of C, H, and N atoms were explicitly treated as valence electrons. Plane-wave function with kinetic

energy less than the energy of 450 eV is included in the basic set. The $1 \times 1 \times 16$ k-point grids based on the strategy of Monkhorst–Pack[67], featuring enough accuracy in the calculation of total energy via convergence test, were used to sample in the Brillouin zone. During the calculations, the convergence value was set to $1.0 \times 10^{-5}$ eV for self-consistent field calculations, and the geometrical optimization will keep running until <0.02 eV/Å of Hellmann–Feynman force per atom.

## Photocatalytic experiments

The photocatalytic $H_2$ evolution experiments were carried out in a 50 mL flask with stirring at room temperature using a 300 W Xe lamp equipped with a UV cut-off filter ($\lambda > 420$ nm). 2.0 mg of the photocatalyst was dispersed in 30 mL of aqueous solution containing 6 mL DMF for dissolution of **N-CNS** and $Na_2S$ and $Na_2SO_3$ as sacrificial reagents. Before irradiation, the solution was bubbled with high-purity nitrogen for 20 min to remove the air.

## Photoelectrochemical measurements

Photoelectrochemical measurements were carried out on a CHI 660e electrochemical workstation in a standard three-electrode electrochemical cell with the photocatalyst-coated FTO as the working electrode, a platinum wire as a counter electrode, and a saturated Ag/AgCl electrode as a reference electrode. A sodium sulfate solution (0.2 M $Na_2SO_4$) was used as the electrolyte. 1 mg **N-CNS** powder was mixed with 1.0 mL acetonitrile and 50 μL Nafion and sonicated for 10 min. The working electrodes were prepared by dropping the suspension (50 μL) evenly onto the surface of the FTO plates and dried at room temperature. The Mott–Schottky plots were measured at the frequencies of 1500, 2000, and 2500 Hz.

## Data availability

Materials and methods, experimental procedures, useful information, characterization studies, $^1H$ NMR spectra, $^{13}C$ NMR spectra, and mass spectrometry data are available in the Supplementary Information. Additional data that support the findings of this study are available from the corresponding author upon request.

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

## Acknowledgements

This work was financially supported by the National Natural Science Foundation of China (22225108, 21971229, 52172053, 22075250, 22122505, 21771161) and the National Key Research and Development Program of China (2022YFE0113800).

## Author contributions

P.D. conceived and designed this research with the help of M.C. and Y. Zhou; Y. Zhou and S.W. synthesized the **N-CNS** material and conducted all characterizations, and EPR measurements. G.S. and Y. Zhu acquired low-dose high-resolution iDPC-STEM images. G.Z. did all the calculation studies. Y. Zhou, M.C., X.Z., and S.W. carried out photophysical studies. Y. Zhou did FT-IR, $^{13}C$ NMR measurements, and photocatalytic reactions. Y. Zhou, M.C., S.W., X.Z., Y. Zhu, G.S., G.Z., and P.D. co-wrote the paper, and all the authors commented on it.

## Competing interests

The authors declare no competing interests.
