## [Peer Review File · Nature Communications]

REVIEWER COMMENTS

Reviewer #1 (Remarks to the Author):

Du and Zhu et al. report the first successful synthesis of large nitrogen-doped carbon nanosolenoids (N-CNS) by a bottom-up approach using Pd-mediated Suzuki-Miyaura coupling and cyclodehydrogenase via the Scholl reaction. The successful synthesis of N-CNS was fully confirmed beyond doubt by GPC, FTIR, solid-state ^{13}C NMR, and Raman techniques. Notably, the authors have also clearly resolved the molecular structure of the single chain intrinsic to the helix of N-CNS by low-dose-integral differential phase-contrast scanning transmission electron microscopy (iDPC-STEM). The unique photophysical properties of N-CNS have also been investigated using UV-Vis, fluorescence, and TRPL spectroscopy. Furthermore, it is shown that the successfully synthesized N-CNS can be used as a highly efficient photocatalyst free of noble metals.

The paper is carefully written and the structure of the N-CNS, including STEM, is well-determined. Therefore, I consider this paper worthy of publication in Nature Communication after the following corrections.

(1) The synthesis in Figure 2 is very confusing, and I think the general reader would have a better understanding of the synthesis if the partial structure is shown and the part changed in the conversion from P1 to N-CNS is indicated, as in Figure 2 of the previous paper (Nature Communication 2022, 13, 1239).

(2) It is stated that N-CNS was performed in a new cycle after three steps of centrifugation, separation, and washing after the reaction, but the experimental procedure is not shown in the SI. It also states that no significant decrease in photocatalytic activity is observed after 12 hours x 10 cycles, but the experimental results are not shown. These should be indicated in the SI.

Reviewer #2 (Remarks to the Author):

The manuscript "A Metal-Free Photoactive Nitrogen-Doped Carbon Nanosolenoid with Broad Absorption in Visible Region for Efficient Photocatalysis" requires certain modifications prior to publication. I have a few suggestions regarding the improvement of the quality of overall manuscript. The manuscript requires major revision.

Below are my comments/ suggestions:

1. Nothing has been described regarding the photocatalysis part in introduction like why is this application important and its relevance in the present work
2. There are many recent reports on photocatalysis which authors can follow: Chemosphere 307 (2022) 135973; J. Clean. Prod. 280 (2021) 124525
3. 'Metal-free' has been emphasized in the title but it's importance has not been described in the text anywhere.
4. The need for large-scale fabrication of nitrogen-doped carbon nanosolenoid (N-CNS) heterojunction materials with Riemann surfaces is mentioned, but there is no explanation of the scientific or practical significance of these materials or their potential applications.
5. Authors can carry out comparison with other materials reported in literature for hydrogen evolution.
6. Conducting any experiment one time is not sufficient. Authors should have carried the study in triplicates and error bars must be used.
7. The efficiency of prepared catalyst must be compared to commercial photocatalyst TiO₂ P25 for hydrogen evolution.

Reviewer #3 (Remarks to the Author):

The Authors reported the synthesis of a nitrogen-doped carbon nanosolenoid (N-CNS) using bottom-up approach with a well-defined structure. The successful synthesis of N-CNS was characterized by GPC, FTIR, solid-state ¹³C NMR, Raman, and iDPC-STEM techniques. In addition, Its photophysical properties were also investigated using UV-Vis, fluorescence, TRPL spectroscopy.

Although the manuscript is well written and the authors characterized the N-CNS using various analytical techniques, the scientific impact of their findings is not high enough to be published in Nature Communications. The authors previously reported the synthesis of CNS in Nature Communications 2022, 13, 1239, as they cited it as Ref. 37. Their current work is about the N-doping of CNS and the photophysical properties. It would be suitable in a more specialized journal.

So, I would not recommend to publish the manuscript in Nature Communications.

Dear Reviewers,

Thanks a lot for your comments and suggestions! These suggestions are constructive and very helpful for revising and improving our manuscript. We have studied the comments carefully and made corrections accordingly. Please find below our point-by-point response.

Thanks again.

Pingwu Du

Point-by-point response to reviewer's comments:

Reviewer #1 (Remarks to the Author):

Du and Zhu et al. report the first successful synthesis of large nitrogen-doped carbon nanosolenoids (N-CNS) by a bottom-up approach using Pd-mediated Suzuki-Miyaura coupling and cyclodehydrogenase via the Scholl reaction. The successful synthesis of N-CNS was fully confirmed beyond doubt by GPC, FTIR, solid-state ^{13}C NMR, and Raman techniques. Notably, the authors have also clearly resolved the molecular structure of the single chain intrinsic to the helix of N-CNS by low-dose-integral differential phase-contrast scanning transmission electron microscopy (iDPC-STEM). The unique photophysical properties of N-CNS have also been investigated using UV-Vis, fluorescence, and TRPL spectroscopy. Furthermore, it is shown that the successfully synthesized N-CNS can be used as a highly efficient photocatalyst free of noble metals.

The paper is carefully written and the structure of the N-CNS, including STEM, is well-determined. Therefore, I consider this paper worthy of publication in Nature Communication after the following corrections.

-We thank this reviewer very much for his/her good comments and suggestions to help us improve our manuscript.

(1) The synthesis in Figure 2 is very confusing, and I think the general reader would have a better understanding of the synthesis if the partial structure is shown and the part changed in the conversion from P1 to N-CNS is indicated, as in Figure 2 of the previous paper (Nature Communication 2022, 13, 1239).

Response: Thank the reviewer for this good comment. As suggested, we have revised Figure 2 (Page 7). The new figure points out the parts that changed during the conversion process from P1 to N-CNS, which can help the general reader understand the synthesis.

Fig. 2 Synthetic approach to N-CNS. Reagents and conditions: (i) **M1** (1.0 equiv.), **M2** (1.0 equiv.), K_2CO_3 (10 equiv.), Aliquat 336 (5 mol%), $\text{Pd}(\text{PPh}_3)_4$ (10 mol%), Ar, toluene/ H_2O (v/v, 5:1), 110 °C, 60 hours; (ii) **P1** (1.0 equiv.), DDQ (16 equiv.), TfOH, Ar, anhydrous CH_2Cl_2 , 0 °C, 24 hours.

(2) It is stated that N-CNS was performed in a new cycle after three steps of centrifugation, separation, and washing after the reaction, but the experimental procedure is not shown in the SI. It also states that no significant decrease in photocatalytic activity is observed after 12 hours x 10 cycles, but the experimental results are not shown. These should be indicated in the SI.

Response: We thank this reviewer very much for these helpful comments. We have added the experimental procedure details in SI. The revisions in SI are as follows:

“... After reaction, the **N-CNS** separated by centrifugation, washed with DCM ... to afford the desired **1b** as a yellow oil in a yield of 95% ...”; (Page S5)

“...The recycling process of **N-CNS** photocatalyst for aza-Henry reaction: To a flame-dried 10 mL vial equipped with a magnetic stir bar with **1a** (104.6 mg, 0.5 mmol) and recycling **N-CNS** (1.5 mg). Nitromethane as the solvent (3 mL) was transferred into the vial via syringe under oxygen and stirred under a blue LED lamp (9 W) for 5 h. Then, the **N-CNS** catalyst was recovered for another cycle. After a few new cycles, the product **1b** can be still obtained in a similar yield...”. (Page S5)

We also added the cycle stability data of 12 hours \times 10 cycles for photocatalytic aza-Henry reactions. The added experimental results are shown in Figure S22 in SI and the caption is described as follows: “Cycle stability of photocatalytic aza-Henry reactions for **N-CNS** sample”. (Page S26)

Figure S22. Cycle stability of photocatalytic aza-Henry reactions for N-CNS sample. Reaction conditions: 2-phenyl-1,2,3,4-tetrahydroisoquinoline **1a** (0.5 mmol, 104.6 mg), photocatalyst (1.5 mg), CH_3NO_2 (3 mL), O_2 (1 atm), LED lamp (9 W).

Overall, we appreciate the reviewers very much for these comments and suggestions to help us to improve our manuscript.

Reviewer #2 (Remarks to the Author):

The manuscript “A Metal-Free Photoactive Nitrogen-Doped Carbon Nanosolenoid with Broad Absorption in Visible Region for Efficient Photocatalysis” requires certain modifications prior to publication. I have a few suggestions regarding the improvement of the quality of overall manuscript. The manuscript requires major revision.

Below are my comments/suggestions:

-We thank this reviewer very much for his/her good comments and suggestions to help us improve our manuscript.

1. Nothing has been described regarding the photocatalysis part in introduction like why is this application important and its relevance in the present work.

Response: We thank the reviewer for this good suggestion. The relevant revisions have been made in the introduction to emphasize the importance of photocatalysis and the relevance in the present work. The revised introduction is as follow:

“...Furthermore, the broad absorption of N-CNS in the visible region enable it as an excellent candidate for photocatalysis. The photocatalysis pathway for solar energy conversion helps to meet the growing demand for energy issues and environmental restoration.^{38,39} Currently, efficient and environmentally friendly photocatalysts are highly demanded for many important energy conversion applications, such as solar water splitting, photocatalytic organic reactions, and so on.⁴⁰⁻⁴² In addition, metal-free catalysts are also of great significance in green chemistry and attract much attention in research.^{43,44} N-CNS, which does not contain any metal component and has photocatalytic properties under visible light, can be used as a potential metal-free catalyst for both visible light-driven H₂ production and efficient photocatalytic organic transformations...”; (in Pages 5-6)

2. There are many recent reports on photocatalysis which authors can follow: Chemosphere 307 (2022) 135973; J. Clean. Prod. 280 (2021) 124525.

Response: Thank you for your helpful comments. We have carefully read these literatures, which are very helpful for improving our manuscript. Now, these related references have been cited and discussed in the revised manuscript. Please see the references 38 and 39 in the revised manuscript.

3. ‘Metal-free’ has been emphasized in the title but it’s importance has not been described in the text anywhere.

Response: Thank this reviewer for this good comment to improve our manuscript. The relevant revisions have been made in the introduction to highlight the importance of “Metal-free”. The revised introduction is as follows (in Pages 5~6):

“...In addition, metal-free catalysts are also of great significance in green chemistry and attract much attention in research.^{43,44} Metal-free catalysts have many advantages that include low-cost and ready availability, low toxicity, higher stability in air and water, and increased synthetic efficiency due to the avoidance of the time-consuming removal of toxic metal traces. N-CNS, which does not contain any metal component and has photocatalytic properties under visible light, can be used as a potential metal-free catalyst for both visible light-driven H₂ production and efficient photocatalytic organic transformations...”;

Besides these, we also have added the word “metal-free material” in some places in our manuscript to emphasize it.

4. The need for large-scale fabrication of nitrogen-doped carbon nanosolenoid (N-CNS) heterojunction materials with Riemann surfaces is mentioned, but there is no explanation of the scientific or practical significance of these materials or their potential applications.

Response: We thank the reviewer for this good suggestion. The relevant revisions have been made in the introduction to explain the scientific significance and potential applications of materials with Riemann surfaces. The added explanations for the scientific significance of the materials with Riemann surfaces in the manuscript are as follows:

“...Riemann surfaces are curved helical structures in chemistry, described as deformed versions of the complex plane in mathematics. Carbon nanosolenoids can be considered to follow a Riemann surface, in which one atomic carbon layer continuously spirals around the line perpendicular to the basal plane...”. (in Page 3)

The added explanations for potential applications of Riemann surfaces materials are as follows:

“...Nanostructured graphitic carbon materials resembling a Riemann surface with helicoid topology are predicted to have interesting magnetic, electrical, and photonic properties. For example, in 2016, Yakobson theoretically predicted a carbon solenoid with Riemann surfaces. This material could emerge a large magnetic field when a voltage is applied and become a brand new magnetic material. The following experimental results proved that such as carbon nanosolenoid material has magnetic properties. Based on all these previous results, we speculate that nitrogen-doped carbon nanosolenoid (N-CNS) heterojunction material with Riemann surfaces will possess special optoelectronic and photocatalytic properties, which is unexplored in this field...”. (in Page 4)

5. Authors can carry out comparison with other materials reported in literature for hydrogen evolution.

Response: Thank this reviewer for this comment. As suggested, we added the comparison with other metal-free materials reported in the literature for hydrogen evolution in SI (Table S3, Page S31). And we revised our manuscript as follows in page 16:

“...We also compared N-CNS with other metal-free materials reported in the literature for hydrogen evolution. These results are showed in Table S3. Most of these materials exhibited low H₂ evolution rates, and only a few materials demonstrated the rates higher than 100 μmol g⁻¹ h⁻¹. In contrast, the hydrogen evolution rate of N-CNS is 190 μmol g⁻¹ h⁻¹. This result demonstrated that N-CNS has good photocatalytic properties...”. (in Page 16)

Table S3. H₂ evolution rate of N-CNS and other materials reported in literature.

Photocatalyst	H ₂ (μmol g ⁻¹ h ⁻¹)	Solution	References
BP/CN	427	Methanol/H ₂ O	S13
Procyanidin–methoxy-benzaldehyde (PC-MB)	252.02	H ₂ O	S14
Carbon-doped BN nanosheets	80	TEOA/H ₂ O	S15
Carbon dots/g-C ₃ N ₄	27.6	TEOA/H ₂ O	S16
P-doped graphene	12	Methanol/H ₂ O	S17
N ₂ -COF	1.7	ACN/TEOA/H ₂ O	S18
Phenyl-triazine oligomers	121	pH 7, 0.5 M, phosphate buffer/TEOA/H ₂ O	S19
N-CNS	190	0.75 M Na₂S, 1.05 M Na₂SO₃/H₂O	This work

6. Conducting any experiment one time is not sufficient. Authors should have carried the study in triplicates and error bars must be used.

Response: We thank this reviewer very much for these helpful comments. As suggested, we did the experiments more times, and added the error bars in Figure 7 (Page 17), Figure S9 (Page S13) and Figure S10 (Page S14).

Fig. 7 Photocatalytic H₂ production by N-CNS. **a** Schematic illustration of photocatalytic H₂ production by N-CNS. **b** H₂ evolution rate of N-CNS photocatalyst at different concentrations of the hole scavenger under visible light. **c** UV-Vis absorption spectrum of N-CNS in DMF with calculated optical bandgap. **d** Cycling runs of N-CNS for photocatalytic H₂ evolution in the presence of 1.0 mg N-CNS photocatalyst in a 30 mL aqueous solution containing 0.75 M Na₂S and 1.05 M Na₂SO₃.

Figure S9. H₂ evolution rates of N-CNS photocatalyst using different sacrificial electron donors (20 vol% TEOA; 0.75 M Na₂S and 1.05 M Na₂SO₃; 0.5 M ascorbic acid; 10 vol% lactic acid; 20 vol% MeOH; 20 vol% MeOH and 20 vol% TEA).

Figure S10. H₂ evolution rates under visible light and full light irradiation.

7. The efficiency of prepared catalyst must be compared to commercial photocatalyst TiO₂ P25 for hydrogen evolution.

Response: We thank the reviewer for this good suggestion. As suggested, we added the comparison with commercial photocatalyst TiO₂ P25 for hydrogen evolution in SI (Table S2, Page S30). The condition is “0.75 M Na₂S, 1.05 M Na₂SO₃/H₂O under visible light”, which is the same condition as our H₂ evolution experiments. And we also revised our manuscript as follows:

“...Compared to commercial photocatalyst TiO₂ P25, the H₂ evolution rate of TiO₂ P25 was only 5.5 μmol g⁻¹ h⁻¹ under the same conditions upon visible light irradiation (Table S2), indicating that N-CNS has better photocatalytic activity than commercial photocatalyst TiO₂ P25 under visible light using the same reaction condition...”. (in Page 16)

Table S2. H₂ evolution rate of N-CNS and TiO₂ P25 under visible light ($\lambda > 420$ nm).

Photocatalyst	H ₂ (μmol g ⁻¹ h ⁻¹)	Solution
N-CNS	190	0.75 M Na ₂ S, 1.05 M Na ₂ SO ₃ /H ₂ O
TiO ₂ P25	5.5	

Overall, we appreciate the reviewers very much for these comments and suggestions to help us to improve our manuscript.

Reviewer #3 (Remarks to the Author):

The Authors reported the synthesis of a nitrogen-doped carbon nanosolenoid (N-CNS) using bottom-up approach with a well-defined structure. The successful synthesis of N-CNS was characterized by GPC, FTIR, solid-state ^{13}C NMR, Raman, and iDPC-STEM techniques. In addition, Its photophysical properties were also investigated using UV-Vis, fluorescence, TRPL spectroscopy.

Although the manuscript is well written and the authors characterized the N-CNS using various analytical techniques, the scientific impact of their findings is not high enough to be published in Nature Communications. The authors previously reported the synthesis of CNS in Nature Communications 2022, 13, 1239, as they cited it as Ref. 37. Their current work is about the N-doping of CNS and the photophysical properties. It would be suitable in a more specialized journal.

So, I would not recommend to publish the manuscript in Nature Communications.

Response: We thank this reviewer very much for her/his time on reviewing our manuscript with critical comments and helpful suggestions. However, we respectfully disagree with the Reviewer based on the following points:

First, nitrogen-doped carbon nanosolenoid (N-CNS) heterojunction material is completely different from the previous work. The previous work only focused on how to synthesize carbon nanosolenoid and characterizations. In this present work, we synthesized a novel nitrogen-doped carbon nanosolenoid (N-CNS) heterojunction material with new photocatalytic properties. In this paper, we also emphasized the importance and effect of doped N atoms on the properties of this curved carbon material: "...The electronic properties of GNRs could also **be tuned by doping with nitrogen**, which has been studied both theoretically and experimentally..." (in Page 4). The results also demonstrated that this present N-CNS material has significantly different physical properties from non-doped nanosolenoid. For example, the UV-Vis spectrum of showed a broad redshifted absorption band in the range of 260-800 nm maximized at ~500 nm (For CNS, absorption in 260-660 nm maximized at ~480 nm) and N-CNS exhibits a broad emission band at 440-820 nm maximized at 564 nm and 661 nm. Compared to previous CNS, the N-CNS has much broader absorption band in the visible light region. This property gives it unique optical properties, such as photocatalytic properties, that were not available in previous CNS material. Based on these, the nitrogen-doped carbon nanosolenoid heterojunction materials are completely new compound and show novel photocatalytic properties.

Second, the design and synthesis of nitrogen-doped carbon nanosolenoid (N-CNS) heterojunction material is meaningful and exhibits many potential applications. We'd like to specifically highlight some unique/novel properties, which were not available in the previous work, as:

(1) This nitrogen-doped carbon nanosolenoid (N-CNS) heterojunction material with Riemann surfaces was proven to possess excellent photocatalytic properties for H_2

production under visible light. (in Page 5)

In further experiments, we added the H₂ evolution comparison results of N-CNS, commercial photocatalyst TiO₂ P25, and other materials in published literatures (Table S2 and S3). These results further support the above conclusion.

(2) N-CNS has efficient catalytic properties in organic photocatalytic reactions. The obtained N-CNS is highly active in catalyzing light-driven aza-Henry reactions between nitromethane and 2-phenyl-1,2,3,4-tetrahydroisoquinoline *via* intermolecular C-H functionalization. More other organic photocatalytic reactions, such as the benzylamine homocoupling and oxygenation of methyl(phenyl)sulfane reactions, were also explored (in Page 6). All these photocatalytic reactions also showed excellent reaction yields and durability.

(3) N-CNS, which does not contain any metal components and has photocatalytic properties under visible light, can be used as a potential metal-free catalyst (in Page 5). Metal-free catalysis is of great significance for environment-friendly and green chemistry.

Therefore, our work is sufficiently novel and has the merits to be accepted for publication in *Nat. Commun.* We hope this reviewer will agree with our novelties.

Overall, we appreciate the reviewers very much for these comments and suggestions to help us to improve our manuscript.